# Cognitive Decline in Glioblastoma (GB) Patients with Different Treatment Modalities and Insights on Untreated Cases

**DOI:** 10.3390/curroncol32030152

**Published:** 2025-03-06

**Authors:** Keyvan Ghadimi, Imane Abbas, Alireza Karandish, Celina Crisman, Emad N. Eskandar, Andrew J. Kobets

**Affiliations:** Department of Neurological Surgery, Albert Einstein College of Medicine, Montefiore Medical Center, Bronx, NY 10461, USAalireza.karandish@einsteinmed.edu (A.K.); akobets@montefiore.org (A.J.K.)

**Keywords:** glioblastoma, cognitive decline, surgery, chemotherapy, radiotherapy, quality of life

## Abstract

Background: Cognitive decline is common in patients with Glioblastoma (GB), occurring in both treated and untreated cases. It frequently presents as impairments in memory, attention, language, or other cognitive functions. In addition, these cognitive deficits can affect quality of life, functional independence, and overall survival, and they are associated with psychiatric conditions such as anxiety and depression. Methods: This narrative review evaluates cognitive deficits in GB patients, both with and without treatment. It also explores the impact of tumor features such as size, location, and histology, along with patient characteristics such as age and education, and discusses the effects of standard therapies, such as surgery, chemotherapy, and radiotherapy, on cognitive outcomes. Results: Cognitive impairment in GB is influenced by tumor- and patient-specific factors, as well as treatment modalities. Initially, combination therapies such as surgery, radiotherapy, and chemotherapy may improve cognitive domains by reducing tumor burden, relieving cerebral edema, and reducing mass effects, subsequently bringing indirect effects of improved mental health and mood. While certain treatments like radiotherapy and chemotherapy carry risks of delayed neurotoxicity, studies indicate that, on balance, treated patients generally show better preservation or improvement in cognitive function than those who go untreated. However, excessive treatment aggressiveness and cumulative neurotoxic effects may diminish cognitive benefits. Conclusion: Cognitive function is an independent factor in GB, which could affect survival in GB patients, therefore making routine cognitive assessments essential for prognosis, treatment planning, and rehabilitation. Neuroprotective agents, cognitive rehabilitation, and personalized, multidisciplinary strategies can help optimize both survival and cognitive preservation.

## 1. Introduction

Glioblastoma (GB) is the most common primary brain tumor in adults, accounting for approximately 47.7% of all primary brain tumors [1]. GB has an incidence of about 3.21 per 100,000 people, is more common in males, and has a median onset age of around 64 years [2]. Because GB is known for rapid growth and poor prognosis, it often requires a combination of aggressive interventions (e.g., surgery, radiotherapy), which may in turn affect the patient’s cognitive abilities [3].

GB is classified by the World Health Organization (WHO) as a grade 4 astrocytoma, given its rapid cellular proliferation, diffuse infiltration, robust angiogenesis, and necrosis regions [3,4,5]. It remains a major clinical challenge due to its invasive nature, tumor heterogeneity, and resistance to available therapies [6].

### 1.1. Pathophysiology

The exact etiology of GB remains incompletely understood, though known risk factors include TP53 mutation, EGFR amplification, PTEN deletion, exposure to ionizing radiation, and various hereditary syndromes [7].

GB is characterized by rapid growth, high invasiveness, and marked genetic heterogeneity [8,9,10]. Molecular profiling differentiates IDH-wildtype grade 4 astrocytomas (known as glioblastomas) from IDH-mutant, WHO grade 4 astrocytomas, which were historically labeled as ‘secondary glioblastomas’. These tumors often harbor IDH1 mutations or EGFR overexpression, contributing to unchecked proliferation and apoptosis resistance [10,11]. Under the 2021 WHO classification, IDH-mutant tumors are now classified as ‘astrocytoma, IDH-mutant, WHO grade 4’ rather than ‘glioblastoma’, although prior studies frequently group them under the broader term glioblastoma.

### 1.2. Prognosis and Survival Rates

Patients with GB generally face a poor prognosis: the median survival is around three months without treatment [12]. With the current standard of care—maximal safe surgical resection (MSR) plus chemoradiotherapy (e.g., Temozolomide)—the median overall survival can increase to roughly 14.6 months [13]. Despite aggressive therapy, the five-year survival rate remains under 5% [14]. Owing to its infiltrative behavior, complete surgical excision is nearly impossible, and the blood–brain barrier (BBB) further limits the efficacy of systemic chemotherapy [15]. Recurrence is common, typically arising near the original tumor site [16].

### 1.3. Aim and Objectives

A cognitive deficit is a significant impairment in mental abilities such as memory, attention, reasoning, language processing, or executive function that often hinders daily functioning and reduces the quality of life [17].

Notably, cognitive decline is common among patients with GB before treatment and can affect up to 90% of these patients [18,19]. Such deficits pose significant challenges for both clinicians and patients, making accurate diagnosis more difficult while reducing the quality of life, functional independence, and decision-making capacity [20,21,22].

Therefore, these cognitive impairments strongly affect lifestyle and overall well-being; particularly in the context of a brain tumor, it is crucial to investigate whether treatment exacerbates or mitigates such deficits. Accordingly, this review examines cognitive outcomes in GB patients who receive various therapeutic interventions compared to those who remain untreated.

## 2. Materials and Methods

We conducted a narrative review evaluating the cognitive outcomes of GB patients before and after treatment. We searched PubMed, MEDLINE, and Web of Science from 2000 to 2024 using the keywords ‘glioblastoma’, ‘cognitive function’, ‘treatment effects’, and ‘quality of life’. Boolean operators (AND, OR) were applied to refine and optimize these search results.

Inclusion criteria included studies of adult patients (≥18 years at diagnosis) with GB that reported cognitive outcomes before and after treatment (surgery, chemotherapy, radiotherapy), or in untreated patients.

Exclusion criteria were pediatric patients (<18 years), absence of cognitive assessments or lack of essential data, and non-English publication.

We extracted data on study design, patient demographics, diagnostic data, treatment modalities, cognitive assessment methods, primary outcomes, tumor characteristics (e.g., size, location), and patient-specific variables (e.g., age, education).

Quality assessment of articles was performed using standardized tools (e.g., CONSORT guidelines) based on the study design, and discrepancies in inclusion criteria were resolved by discussion among reviewers.

## 3. Cognitive Deficits

### 3.1. Prevalence and Nature of Cognitive Deficits

Research indicates that GB-related cognitive deficits can affect various domains such as memory, attention, executive functions, language, and visuospatial skills [23,24]. Tucha et al. [24] found that many GB patients already manifest difficulties in attention, memory, and organization before treatment. Likewise, a systematic review by Acevedo-Vergara et al. [17] highlighted that high-grade gliomas often cause substantial cognitive changes in language, attention, memory, empathy, and executive functions (Table 1).

### 3.2. Impact of Tumor Location on Cognition

Tumor location strongly influences the type and severity of cognitive impairments in GB. Lesions in the frontal or prefrontal lobes may cause deficits in executive functions, such as planning, decision-making, and inhibitory control [22,27]. In untreated patients, frontal tumors often lead to marked executive dysfunction as mass effect and tumor growth proceed unchecked. In contrast, treated patients may fare better, with milder impairments, particularly when surgical resection spares critical areas [28].

Temporal lobe involvement often results in memory impairment and language comprehension issues [20]. Untreated temporal lobe lesions can rapidly degrade cognitive function, especially in verbal memory. Treated patients, however, may retain more capabilities, aided by compensatory neuroplasticity and radiotherapy-induced tumor control [20].

Parietal and occipital lobe tumors can compromise spatial orientation, attention, and visual processing [29]. Untreated disease in these regions can lead to neglect syndromes or cortical blindness, whereas treatment may slow the decline or partially preserve function, due to neurorehabilitation efforts [24]. Additionally, subcortical white matter involvement disrupts neural networks, often causing broad deficits in processing speed and connectivity [30].

### 3.3. Factors Influencing Cognitive Function

Multiple factors contribute to cognitive impairments in GB, including the following:•Tumor Characteristics: Tumor size, histology, WHO grade, and location correlate strongly with cognitive function [22,31,32]. Larger, more infiltrative tumors worsen outcomes for untreated patients due to a greater mass effect and disruption of surrounding tissue. In contrast, patients with smaller residual tumors post-resection tend to perform better cognitively [24]. Also, IDH-mutant, WHO grade 4 astrocytoma (formerly called secondary GB) often portends longer survival and better cognitive outcomes, irrespective of treatment [33].•Patient Characteristics: Variables such as age, education level, and IDH status influence the extent of cognitive deficits [22,34,35]. Younger treated patients typically display better cognitive resilience than untreated individuals, who may show swifter, more global declines [36].•Treatment Modalities: Maximal safe resection (MSR) remains essential to GB management and is linked to better cognitive outcomes than subtotal or no resection. MSR can reduce tumor burden, ease mass effect, and lower edema, but it carries a risk of damaging healthy tissue in eloquent areas [37,38].•Radiotherapy is standard adjuvant therapy but can cause delayed neurotoxicity, including hippocampus damage that impairs memory [13,39]. Chemotherapy (particularly Temozolomide) boosts survival but may also cause executive and memory deficits [13]. Combination therapy often yields the best survival gains, although cumulative neurotoxic effects may undermine cognition [40,41].•Medication Use: Some medications improve cognition by reducing tumor-related symptoms; others can exacerbate deficits [42,43]. Antiepileptic drugs (AEDs) and corticosteroids can help manage edema and seizures, temporarily improving cognition. Older AEDs (e.g., phenytoin) cause sedation and memory problems, while long-term corticosteroid use can produce mood changes and memory impairment [44,45].•Clinical Symptoms: Depression, anxiety, and neuropsychiatric complications can worsen cognitive function [26,46]. Untreated patients often experience intense psychological distress that magnifies cognitive deficits, whereas treated patients may benefit from better mood and mental health [20,47].

## 4. Cognitive Impairment and Survival

Cognitive function contributes significantly as a dependent factor to quality of life and independently predicts survival in GB [48,49]. Multiple studies have demonstrated that poorer cognitive performance is linked to reduced survival, independent of other clinical factors [50,51]. Evidence increasingly underscores cognitive function’s prognostic significance, as it often reflects tumor burden, treatment complications, or disease progression. For instance, Meyers et al. observed that verbal memory correlated strongly with survival in recurrent malignant glioma [48]. Johnson et al. showed that early deficits in attention and executive functions predict survival in newly diagnosed GB [35]. Similarly, Klein et al. reported decreased survival in high-grade glioma patients with cognitive impairment, irrespective of therapy (relative risk: 4.099) [50]. These findings suggest that routine baseline and follow-up cognitive assessment can guide therapeutic intensity and balance aggressive treatment with functional preservation.

Hence, cognitive assessments should be integral to GB management, aiding prognosis, guiding therapy, and influencing overall outcomes [15,35].

## 5. Follow-Up Timing for Patients with GB After Treatment Options

An important component of post-treatment management of GB is timely follow-up to detect cognitive deficits. Current data suggest that impairment may develop within ~3 months after surgery, underlining the need for early cognitive evaluation, as it can be an independent predictor of survival [52]. Because GB typically requires multimodal therapy (surgery, chemotherapy, radiotherapy), follow-up scheduling can be complex. Radiotherapy and chemotherapy both carry a risk of long-term cognitive impairments. Studies recommend follow-up assessments of 6 to 12 months for patients receiving combined modalities [53,54]. Regarding the frequency of evaluations, both immediate and delayed cognitive changes can occur. Thus, monthly check-ups during the first three months are advisable to monitor early decline, followed by evaluations every three months for the next year [52,54].

## 6. Treatment Options and Their Cognitive Effects

The main goals of GB therapy are prolonging survival, alleviating symptoms, and sustaining or enhancing neurological function via surgery, chemotherapy, radiotherapy, or combined approaches [15]. Although such interventions can improve cognition by reducing mass effects and limiting progression, they also carry risks of further cognitive decline [55].

### 6.1. Surgical Interventions

#### 6.1.1. Benefits and Risks

MSR is central to GB management [56], often improving cognition by lowering intracranial pressure and mass effect [34]. However, surgery risks harming healthy tissue, leading to postoperative edema and anesthesia-related complications, any of which can worsen cognitive declines [36,57,58].

#### 6.1.2. Techniques to Preserve Cognition

Neurosurgeons employ various techniques to optimize resection while safeguarding cognition. Awake craniotomy enables real-time monitoring of language and motor areas, reducing postoperative deficits [59,60]. Intraoperative brain mapping identifies critical brain areas to avoid [61], and neuronavigation enhances surgical precision [62]. Though these methods often improve survival and preserve neurological function, some risk of cognitive decline remains [37].

### 6.2. Chemotherapy

#### 6.2.1. Common Agents Used

Temozolomide, the standard chemotherapeutic agent for GB, is typically given alongside radiotherapy [63]. It crosses the BBB and induces DNA damage in tumor cells [64]. Nevertheless, the BBB still limits drug penetration into the cerebrospinal fluid (CSF), promoting investigations into BBB destruction techniques to enhance delivery [64].

#### 6.2.2. Neurotoxic Effects

Temozolomide and related agents improve overall and progression-free survival but can trigger fatigue, concentration difficulties, and other cognitive impairments [65,66]. As an alkylating agent, Temozolomide methylates DNA to kill rapidly dividing tumor cells [64], yet it may also affect non-dividing or slow-dividing cells (including neurons and glia), causing neurotoxicity and cognitive decline [19]. Higher doses correlate with a greater neurotoxic risk [67].

Other chemotherapeutic agents, such as Nitrosoureas, have higher neurotoxicity profiles and are therefore less commonly used [68].

### 6.3. Radiotherapy

#### 6.3.1. Radiotherapy Approaches

Radiotherapy, often used post-surgically, targets and destroys residual tumor cells. Techniques include external beam radiotherapy, intensity-modulated radiotherapy, and stereotactic radiosurgery [69,70].

#### 6.3.2. Cognitive Side Effects

Radiation may produce both acute and delayed cognitive impairment via white matter necrosis, vascular injury, and neuroinflammation [71,72,73]. Chronic effects, such as memory impairment, can emerge months or even years post-treatment [74]. Fractionation schedules and hippocampal-sparing techniques aim to mitigate these impacts, yet radiotherapy still carries an inherent neurotoxic risk [75,76].

### 6.4. Combination Therapies

Combining treatments typically yields better survival than monotherapy but may heighten cognitive deficit [36,77]. Therefore, balancing treatment efficacy with quality-of-life goals is critical [78] (Table 2).

### 6.5. Immunotherapy

Approaches such as immune checkpoint inhibitors, vaccines, and CAR T-cell therapy aim to harness the immune system against GB [79]. Early trials show potential for enhanced efficacy with possibly fewer cognitive side effects [80]. However, recent studies indicate that immune effector cell-associated neurotoxicity syndrome (ICANS) and cytokine release syndrome (CRS), observed in hematologic malignancies treated with CAR T-cells, can lead to delayed or mild cognitive deficits [81,82]. While the incidence of severe persistent impairment is relatively low, patients with more severe ICANS may report worse long-term perceived cognition. Further large-scale trials focusing on primary or metastatic brain tumors are needed to assess the true extent of immunotherapy-related cognitive changes [82,83].

**Table 2 curroncol-32-00152-t002:** Treatment modalities for glioblastoma and their cognitive effects.

Treatment Modality	Potential Cognitive Benefits	Potential Cognitive Risks	Strategies to Mitigate Risks	References
**SURGICAL RESECTION**	Reduces mass effect, alleviates symptoms	Risk of damage to eloquent brain areas	Awake craniotomy, intraoperative mapping	[34]
**RADIOTHERAPY**	Controls residual tumor growth	White matter damage, neuroinflammation	Fractionation schedules, hippocampal-sparing techniques	[69,70]
**CHEMOTHERAPY** **(TEMOZOLOMIDE)**	Crosses BBB, prolongs survival	Fatigue, concentration difficulties	Dose management, supportive care	[64]
**COMBINED MODALITY THERAPY**	Increased efficacy against tumor cells	Compounded neurotoxicity	Personalized treatment plans	[36,77]
**EXPERIMENTAL THERAPIES**	Potential for targeted treatment	Unknown long-term cognitive effects	Clinical trials, close monitoring	[79]
**COGNITIVE REHABILITATION**	Improves specific cognitive deficits	Requires sustained patient engagement	Personalized rehabilitation programs	[84]

## 7. Comparative Analysis of Cognitive Outcomes

Comparing cognitive outcomes in treated and untreated GB patients is vital for guiding therapy and counseling. Such comparisons provide insights into how treatment modalities might affect cognitive function over time.

### 7.1. Treated vs. Untreated Patients

Research indicates that treated GB patients frequently experience early stabilization or improvement in cognitive function, attributed to tumor debulking and symptom management [24]. However, neurotoxicity effects from radiation and chemotherapy can contribute to cognitive decline over time, sometimes months or even years after treatment completion [48].

By contrast, in untreated patients, cognitive decline is typically much more rapid and progressive due to the tumor’s aggressive nature, leading to significant tumor growth and quick invasion of the adjacent parenchyma [85]. Acevedo-Vergara et al. reported that high-grade gliomas cause significant alterations in cognitive domains, and patients require neuropsychological evaluation to determine the grade of cognitive dysfunction. Cognitive impairments are generally more severe in high-grade gliomas compared to low-grade gliomas, where brain plasticity processes are faster and more effective, potentially allowing for greater adaptation and less pronounced cognitive difficulties [17]. 

### 7.2. Factors Influencing Outcomes

Age, baseline cognition, tumor size, location (especially frontal lobe), and molecular markers all shape cognitive outcomes in GB. While more extensive resection often improves survival, it may heighten cognitive risks. Additionally, more aggressive therapies can escalate side effects [20,43,56,66]. 

### 7.3. Importance of Neuropsychological Evaluation

Standard neuropsychological assessments are essential in the management of patients with brain tumors before and after surgery to monitor cognitive function, guide rehabilitation programs, and evaluate treatment outcomes [48]. Performing assessments helps identify specific cognitive deficits and inform interventions aimed at improving quality of life [84].

## 8. Mitigation Strategies for Cognitive Decline

Addressing the cognitive decline in GB patients involves a comprehensive approach, including advanced medical techniques and supportive therapies.

### 8.1. Advanced Surgical Techniques

•Awake Craniotomy: Reduces the risk of postoperative cognitive deficits by allowing for real-time monitoring [60,61].•Functional Brain Mapping: Guides surgical planning to minimize cognitive risks [59].•Intraoperative Technologies: Neuronavigation and intraoperative imaging enhance surgical precision [37,62].

### 8.2. Pharmacological Interventions

Research is exploring neuroprotective agents to prevent cognitive decline:•Memantine: Shown to reduce cognitive decline during radiotherapy [86].•Donepezil: May improve cognitive function in irradiated patients [87].

Moreover, a recent systematic review by Kirkman et al. [88] indicates that a variety of pharmacological strategies (e.g., methylphenidate, modafinil, Ginkgo biloba, and Shenqi fuzheng) have been studied in adult brain tumor populations, with some evidence of cognitive benefits. However, many of these studies suffered from methodological limitations or a high risk of bias, and it remains unclear whether improvements persist after cessation of therapy. Future large-scale, randomized trials are needed to confirm durable cognitive gains.

### 8.3. Rehabilitation Programs

#### 8.3.1. Cognitive Rehabilitation

Structured programs aim to improve specific cognitive deficits through targeted exercises and other strategies [84]. Interventions may focus on the following:•Memory Training: Techniques to enhance recall and retention [18].•Attention and Concentration: Exercises to improve focus [89].•Executive Function: Problem-solving tasks and organizational skills [90].

Studies have demonstrated that cognitive rehabilitation can lead to significant improvements in cognitive performance and daily functioning [84]. For instance, Kirkman et al. [88] identified interventions such as general cognitive rehabilitation, working memory training, goal management training, and aerobic exercise as potentially beneficial, though long-term durability remains uncertain.

#### 8.3.2. Multidisciplinary Support

Comprehensive care includes the involvement of occupational therapists, speech-language pathologists, and neuropsychologists [91]. Support groups and counseling services address emotional and psychological needs [92].

## 9. Quality-of-Life Considerations

Quality of life (QoL) is a multifaceted construct encompassing physiological, psychological, social, and functional well-being. These domains exhibit intricate interdependencies and are integral to an individual’s overall health status [93]. In the context of GB, cognitive impairment is a critical factor that significantly affects the patient’s ability to perform daily activities, maintain relationships, and engage in meaningful pursuits, ultimately diminishing their overall QoL [94].

### 9.1. Psychological Impact

Cognitive deficits can lead to depression and anxiety, social isolation, and emotional instability [95]. These issues contribute to social withdrawal and feelings of helplessness, increasing isolation and worsening emotional distress [40].

Providing psychological support, including cognitive–behavioral therapy (CBT) and counseling, is essential to help patients and families cope [96].

### 9.2. Caregiver Burden

Caregivers of patients with GB often experience significant stress, as they must adapt to the disease progression and the patient’s changing needs and behaviors. The burden includes constant monitoring, assistance with daily tasks, and navigating emotional complexities [97]. Support services and respite care can help alleviate the caregiver’s burden by offering practical assistance and providing emotional support, thereby improving the caregiver’s well-being and ensuring long-term care for the patient [96].

## 10. Future Directions and Research Opportunities

Advancements in understanding GB biology and cognitive neuroscience offer hope for improved outcomes.

### 10.1. Novel Therapies

#### 10.1.1. Targeted Molecular Therapies

Drugs targeting specific genetic mutations and pathways in GB cells may improve treatment specificity and reduce toxicity [98]. Examples include the following:•EGFR Inhibitors: Target overexpressed receptors in GB [99].•VEGF Inhibitors: Reduce angiogenesis, e.g., Bevacizumab [100].

#### 10.1.2. Gene Therapy

Gene editing technologies like CRISPR/Cas9 offer avenues for correcting genetic abnormalities in tumor cells [18,101].

### 10.2. Personalized Medicine

Integrating genomic and molecular profiling into clinical practice enables tailored treatments based on individual tumor characteristics [102]. This approach aims to maximize efficacy and minimize adverse effects, including cognitive decline [103].

### 10.3. Neuroprotective Strategies

Ongoing research is focused on identifying agents and interventions that can protect neural tissue during treatment [104]. Investigations into the mechanisms of radiation-induced cognitive decline may lead to novel protective measures [105].

### 10.4. Rehabilitation Innovations

Advancements in neurorehabilitation, such as computerized cognitive training and virtual reality therapies, offer new modalities for cognitive improvement [106]. Research into neuroplasticity may uncover ways to promote brain recovery [107].

### 10.5. Clinical Trials and Collaborative Research

Participation in clinical trials provides access to innovative therapies and contributes to the collective understanding of GB [108]. Collaborative efforts across institutions enhance research quality and accelerate progress [13].

## 11. Conclusions

Available evidence confirms that GB—whether treated or untreated—poses a substantial threat to both survival and neurological function. Cognitive decline remains a major concern, undermining independence, social relationships, and overall well-being. While surgical resection, radiotherapy, and chemotherapy can extend survival and may preserve selected cognitive domains, these modalities also carry risks of neurotoxicity, particularly when used in combination or at higher doses. Similarly, immunotherapies show promise in controlling tumor progression, yet immune-related neurotoxicity can occur and may have longer-term implications for cognition.

Recent systematic reviews highlight that various pharmacological and nonpharmacological interventions can help mitigate cognitive deficits in brain tumor populations, although high-quality evidence is still limited. Likewise, studies examining CAR T-cell therapy in hematological malignancies provide preliminary insights into late cognitive changes and suggest close monitoring for neuropsychiatric symptoms. Therefore, a comprehensive, multidisciplinary approach that integrates tumor-directed treatments with neuroprotective strategies, cognitive rehabilitation, and psychosocial support appears essential to optimize outcomes.

Future research should continue pursuing more effective and less neurotoxic treatments, including newer immunotherapies and neuroprotective agents, while standardizing cognitive assessments to refine prognosis and personalize care. In particular, large-scale, collaborative clinical trials with robust neurocognitive endpoints are necessary to confirm the durability of these interventions and better characterize risk factors for long-term impairment.

## Figures and Tables

**Table 1 curroncol-32-00152-t001:** Cognitive domains affected in glioblastoma patients and influencing factors.

Cognitive Domain	Common Impairments	Influencing Factors	References
**MEMORY**	Short-term memory loss, difficulty recalling	Temporal lobe involvement, tumor size	[24]
**ATTENTION**	Reduced concentration, distractibility	Frontal and parietal lobe involvement	[22]
**EXECUTIVE FUNCTIONS**	Impaired planning, decision-making, inhibitory control	Frontal lobe tumors, patient age, treatment effects	[25]
**LANGUAGE**	Aphasia, word-finding difficulties	Left hemisphere tumors, surgical impact	[22]
**VISUOSPATIAL SKILLS**	Difficulty with spatial orientation and perception	Parietal and occipital lobe involvement	[24]
**PROCESSING SPEED**	Slowed cognitive processing	Treatment effects, overall disease burden	[24]
**EMOTIONAL PROCESSING**	Depression, anxiety, altered affect	Tumor location, corticosteroid use	[26]

## Data Availability

This study did not create or analyze new data, and data sharing does not apply to this article.

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
