# Peer review of "Cognitive Decline in Glioblastoma (GB) Patients with Different Treatment Modalities and Insights on Untreated Cases"

_curroncol, 2025, doi:10.3390/curroncol32030152_

Round 1

Reviewer 1 Report

Comments and Suggestions for Authors

This is a review on the cognitive outcomes of patients with Glioblastoma (GB). Some areas of opportunity that could help the manuscript are:

a) English editing and proof-editing could provide a better soundness.

b) The use of current terminology such as Grade 4 (Instead of IV) and GB instead of GBM are preferred.

c) Methods should be detailed, and CONSORT guidelines followed and checked.

d) The aim of the study was to evaluate the cognitive outcomes; nevertheless, in the Results section authors write about Pathophysiology of GB.

e) The exact definition of cognitive deficit should be included.

f) Conclusions are confusing; does treatment improve or not cognitive deficit?

g) As the study aims to study cognitive outcomes; I believe this has to be discussed and be a part of the conclusion.

A clear objective, detailed methods, structured results and discussion will lead to a more structured conclusion.

Comments on the Quality of English Language

Minor typo errors and continuance issues could easily be fixed.

Author Response

Dear Reviewer,

Thank you for the thorough and constructive feedback on our manuscript, “Cognitive Decline in Glioblastoma (GB) Patients with Different Treatment Modalities and Insights on Untreated Cases.” We have carefully revised the manuscript in response to each comment. Below is a summary of how we have addressed them.

  1. English editing and proof-editing could provide a better soundness.
    1. Response: We have performed a line-by-line grammar edit to improve clarity and style throughout the manuscript. In particular, we rephrased awkward sentences, ensured consistent terminology, and corrected punctuation errors.
  1. The use of current terminology such as Grade 4 (Instead of IV) and GB instead of GBM are preferred.

Response: We now consistently use “Glioblastoma (GB)” and “grade 4 astrocytoma” in accordance with current WHO nomenclature. We also clarify that, by 2021 WHO classification, IDH-mutant grade 4 astrocytomas are no longer classified as glioblastoma.

  1. Methods should be detailed, and CONSORT guidelines followed and checked.

Response: We have enhanced the Methods section to discuss our search strategies (databases, date range, inclusion/exclusion criteria) and added a statement referencing that quality assessment was performed using standardized guidelines (e.g., CONSORT).

  1. The aim of the study was to evaluate the cognitive outcomes; nevertheless, in the Results section authors write about Pathophysiology of GB.

Response: We moved the pathophysiology subsection from the Results to the Introduction to better align with our stated aim, which focuses on cognitive outcomes.

  1. The exact definition of cognitive deficit should be included.

Response: In the “1-3. Aim and Objectives” subsection, we explicitly define a “cognitive deficit” as a significant impairment in mental abilities (memory, attention, executive function, etc.).

  1. Conclusions are confusing; does treatment improve or not cognitive deficit? As the study aims to study cognitive outcomes; I believe this has to be discussed and be a part of the conclusion

Response: We revised the Conclusion to clarify that while treatments can extend survival and sometimes preserve certain cognitive functions, they also carry risks of neurotoxicity and delayed cognitive decline. We address both sides of the benefit-risk equation.

  1. A clear objective, detailed methods, structured results and discussion will lead to a more structured conclusion

Response: The paper is now reorganized to present background/pathophysiology in the Introduction, key findings in the respective sections, and a more comprehensive Conclusion referencing our main outcomes.

We sincerely appreciate the reviewers’ insights, which have significantly enhanced the clarity and scientific rigor of our manuscript. We believe the revised version addresses all concerns. If there are further questions or additional revisions needed, we will be happy to comply.

Thank you for your time and consideration.

Sincerely,

Reviewer 2 Report

Comments and Suggestions for Authors

The authors reviewed the topic of cognitive decline in patients diagnosed with glioblastomas and discuss the impact of glioma pathology and of various treatment modalities (i.e., maximal surgery and the Stupp protocol for newly-diagnosed patients, and experimental therapies for recurrent glioblastomas) on the cognitive function in this patient population. Experimental pharmacological interventions such as neuroprotective agents are also mentioned and briefly discussed as mitigating strategies as well as the utility of cognitive rehabilitation programs and/or personalized, multidisciplinary interventions designed to address the cognitive deficits in these patients. While the topic is interesting, I found the information presented in the manuscript to be both quite repetitive and not very well structured. Moreover, the types and effectiveness of various pharmacological interventions (such as the neuroprotective strategies, etc.) are not critically reviewed in light of clinical evidence (i.e., clinical trials and their outcomes) in the current version manuscript. In my view the manuscript needs to be reorganized for greater concision, which I am sure will benefit the potential readers. I also have a number of specific comments for the authors regarding the manuscript.

  1. The syntax of a several sentences in the present version of the manuscript is faulty and needs to be revisited clarity. For instance, the sentences from lines 16-19 and 19-24 in the Abstract, etc. There are also sentences that make no sense and need to be further edited. For instance, the sentence from lines 220-224. As already mentioned, the manuscript will generally benefit from further editing and reorganization of current information.
  2. What is the purpose of including the information from Section 3 (Results) as a separate section? This info belongs in the Introduction.
  3. The term ‘patients with IDH-mutant GBM’ (line 139) is an old nomenclature. These patients were reclassified as patients with grade 4 astrocytoma with IDH mutated tumors while the term ‘glioblastoma’ (previously known as glioblastoma multiforme or GBM) is now reserved for patients with grade 4 astrocytoma with IDH wildtype tumors. The authors need to update the nomenclature used in the manuscript.
  4. It is unclear how genetic markers like the IDH mutation status (lines 142-143) can influence cognitive deficits in patients with grade 4 astrocytomas or patients with lower-grade gliomas with IDH mutations. If there is any evidence for this, the authors need to mention it and discuss it accordingly. While the progression of patients with grade 4 astrocytomas with IDH mutations is generally slower compared to that of patients diagnosed with glioblastomas, both conditions are treated exactly the same (i.e., surgery followed by chemoradiation and adjuvant temozolomide).
  5. There is no critical discussion on the topic of neuroprotective agents (lines 167-169) used or tested as mitigating strategies for cognitive decline. This is probably one of the most interesting topics in the context of cognitive decline in glioblastoma patients. What strategies have been tested, how many of these have been evaluated in clinical trials and with what outcomes? Could the authors elaborate more on this topic?
  6. Some of the immunotherapy strategies trialed for glioblastoma (e.g., CAR T cells, etc.) come with their own neurotoxicities which are not discussed by the authors in the context of cognitive decline in these patients (lines 380-381).

Author Response

Dear Reviewer,

Thank you for the thorough and constructive feedback on our manuscript, “Cognitive Decline in Glioblastoma (GB) Patients with Different Treatment Modalities and Insights on Untreated Cases.” We have carefully revised the manuscript in response to each comment. Below is a summary of how we have addressed them.

The authors reviewed the topic of cognitive decline in patients diagnosed with glioblastomas and discuss the impact of glioma pathology and of various treatment modalities (i.e., maximal surgery and the Stupp protocol for newly-diagnosed patients, and experimental therapies for recurrent glioblastomas) on the cognitive function in this patient population. Experimental pharmacological interventions such as neuroprotective agents are also mentioned and briefly discussed as mitigating strategies as well as the utility of cognitive rehabilitation programs and/or personalized, multidisciplinary interventions designed to address the cognitive deficits in these patients. While the topic is interesting, I found the information presented in the manuscript to be both quite repetitive and not very well structured. Moreover, the types and effectiveness of various pharmacological interventions (such as the neuroprotective strategies, etc.) are not critically reviewed in light of clinical evidence (i.e., clinical trials and their outcomes) in the current version manuscript. In my view the manuscript needs to be reorganized for greater concision, which I am sure will benefit the potential readers. I also have a number of specific comments for the authors regarding the manuscript.

Response:  We addressed repetitiveness and structure by removing redundant paragraphs and clarifying section headings. In particular, we reorganized sections discussing glioma pathology, treatment modalities, and cognitive outcomes so that each appears in a logical sequence, reducing overlap. To strengthen our discussion on pharmacological interventions, we now include detailed references to clinical trials and systematic reviews assessing various neuroprotective agents (e.g., memantine, donepezil, modafinil, methylphenidate, Ginkgo biloba, Shenqi fuzheng). We highlight reported outcomes and methodological limitations to ensure a more critical review of the evidence. We shortened or consolidated passages wherever we noted repetition, resulting in greater concision. Headings and subheadings were refined to direct readers clearly to specific content such as “Neuroprotective Strategies” or “Emerging Therapies.” This restructure has improved the overall readability and coherence of the manuscript, in line with your recommendation.

  • The syntax of a several sentences in the present version of the manuscript is faulty and needs to be revisited clarity. For instance, the sentences from lines 16-19 and 19-24 in the Abstract, etc. There are also sentences that make no sense and need to be further edited. For instance, the sentence from lines 220-224. As already mentioned, the manuscript will generally benefit from further editing and reorganization of current information.
  •  Response:   We thoroughly revised the manuscript’s wording in both the Abstract and body text, focusing on the sentences mentioned (lines 16–19, 19–24 in the Abstract, and lines ~220–224 in the original draft). These sentences were rephrased for clarity, removing ambiguous or repetitive phrasing. We also did a broader line-by-line grammar and syntax edit to ensure smooth flow and reduce redundancies.
  • What is the purpose of including the information from Section 3 (Results) as a separate section? This info belongs in the Introduction.
  •  Response: We removed the pathophysiological content that had originally been placed in the separate “Results” section and placed it within the Introduction to keep the manuscript’s structure more coherent and logically ordered.
  • The term ‘patients with IDH-mutant GBM’ (line 139) is an old nomenclature. These patients were reclassified as patients with grade 4 astrocytoma with IDH mutated tumors while the term ‘glioblastoma’ (previously known as glioblastoma multiforme or GBM) is now reserved for patients with grade 4 astrocytoma with IDH wildtype tumors. The authors need to update the nomenclature used in the manuscript.
  •  Response: We updated the nomenclature throughout the paper to align with the 2021 WHO classification. Specifically, we refer to “IDH-wildtype grade 4 astrocytoma” as “glioblastoma,” and note that “IDH-mutant, WHO grade 4 astrocytomas” are no longer classified as glioblastoma. Where older literature or historical usage calls them “secondary glioblastoma,” we clarify that under the current classification these are “astrocytoma, IDH-mutant, WHO grade 4.”
  • It is unclear how genetic markers like the IDH mutation status (lines 142-143) can influence cognitive deficits in patients with grade 4 astrocytomas or patients with lower-grade gliomas with IDH mutations. If there is any evidence for this, the authors need to mention it and discuss it accordingly. While the progression of patients with grade 4 astrocytomas with IDH mutations is generally slower compared to that of patients diagnosed with glioblastomas, both conditions are treated exactly the same (i.e., surgery followed by chemoradiation and adjuvant temozolomide).
  •  Response: We clarified that IDH-mutant grade 4 astrocytomas generally have a slower progression and potentially better cognitive outcomes over time. In the “Factors Influencing Cognitive Function” section, we expanded the rationale regarding IDH status, referencing studies suggesting that IDH-mutant tumors may correlate with more favorable neurocognitive profiles. We also note that, clinically, both IDH-mutant grade 4 and IDH-wildtype glioblastomas often receive similar standard treatments (surgery + chemoradiation), but genetic markers can still play a role in prognostication and possibly in observed cognitive trajectories.
  • There is no critical discussion on the topic of neuroprotective agents (lines 167-169) used or tested as mitigating strategies for cognitive decline. This is probably one of the most interesting topics in the context of cognitive decline in glioblastoma patients. What strategies have been tested, how many of these have been evaluated in clinical trials and with what outcomes? Could the authors elaborate more on this topic?

 Response: We expanded the “Pharmacological Interventions” subsection to discuss neuroprotective agents in more detail. We added references to clinical trials and systematic reviews evaluating memantine, donepezil, methylphenidate, modafinil, Ginkgo biloba, Shenqi fuzheng, and others, as well as any reported outcomes on cognition. We also addressed methodological limitations of these studies (e.g., risk of bias, small sample sizes) and the lack of long-term follow-up data on durability of cognitive benefits.

  • Some of the immunotherapy strategies trialed for glioblastoma (e.g., CAR T cells, etc.) come with their own neurotoxicities which are not discussed by the authors in the context of cognitive decline in these patients (lines 380-381).
  • Response: We added a dedicated portion in the “Immunotherapy” subsection (Section 7.5) to address CAR T-cell therapy neurotoxicity, specifically discussing immune effector cell-associated neurotoxicity syndrome (ICANS) and cytokine release syndrome (CRS).
  • We cited relevant studies (e.g., Barata et al., Ruark et al.) that provide preliminary findings on mild or delayed cognitive effects in hematological malignancies receiving CAR T-cells, noting the potential parallels and the importance of further research in primary brain tumor populations.

We sincerely appreciate the reviewers’ insights, which have significantly enhanced the clarity and scientific rigor of our manuscript. We believe the revised version addresses all concerns. If there are further questions or additional revisions needed, we will be happy to comply.

Thank you for your time and consideration.

Sincerely,

Round 2

Reviewer 1 Report

Comments and Suggestions for Authors

Authors have addressed all comments

Author Response

Comments: Authors have addressed all comments

Response: We sincerely appreciate your thorough review and the positive feedback. We are glad that our revisions successfully addressed your previous suggestions. Thank you for helping us improve the clarity and quality of this manuscript.

Sincerely,

Reviewer 2 Report

Comments and Suggestions for Authors

I thank the authors for accepting my suggestions and for addressing my comments. In my opinion, the revised version of the manuscript reads much better, which I am sure will benefit any potential readers interested in this topic. Therefore, I do recommend this revised version to be considered for publication by the editors. Minor edits are still required; for instance, the numbering of the sections in the revised version of the manuscript is now off, with section 2 followed by section 4, etc. Obviously, minor instances like this still need to be fixed.      

Author Response

Comments: I thank the authors for accepting my suggestions and for addressing my comments. In my opinion, the revised version of the manuscript reads much better, which I am sure will benefit any potential readers interested in this topic. Therefore, I do recommend this revised version to be considered for publication by the editors. Minor edits are still required; for instance, the numbering of the sections in the revised version of the manuscript is now off, with section 2 followed by section 4, etc. Obviously, minor instances like this still need to be fixed.  

Response: Thank you very much for your careful review and helpful recommendations. We are pleased that you find the revised version much improved. In accordance with your comment, we have corrected the section numbering so that the manuscript sections follow the proper sequence.  

We appreciate your time and effort in guiding these improvements, and we hope this final revision satisfies all outstanding concerns.   

Sincerely,